# The Synergistic Effect of Baloxavir and Neuraminidase Inhibitors against Influenza Viruses In Vitro

**DOI:** 10.3390/v16091467

**Published:** 2024-09-14

**Authors:** Xiaojia Guo, Lei Zhao, Wei Li, Ruiyuan Cao, Wu Zhong

**Affiliations:** 1National Engineering Research Center for the Emergency Drug, Beijing Institute of Pharmacology and Toxicology, Beijing 100850, China; 15227117791@163.com (X.G.); a_moon1096@163.com (W.L.); 2Beijing Sunho Pharmaceutical Co., Ltd., Beijing 102600, China; leizhao-hit@hotmail.com

**Keywords:** drug combination, baloxavir, neuraminidase inhibitors, influenza virus, drug-resistant strains

## Abstract

Influenza viruses remain a major threat to human health. Four classes of drugs have been approved for the prevention and treatment of influenza infections. Oseltamivir, a neuraminidase inhibitor, is a first-line anti-influenza drug, and baloxavir is part of the newest generation of anti-influenza drugs that targets the viral polymerase. The emergence of drug resistance has reduced the efficacy of established antiviral drugs. Combination therapy is one of the options for controlling drug resistance and enhancing therapeutical efficacies. Here, we evaluate the antiviral effects of baloxavir combined with neuraminidase inhibitors (NAIs) against wild-type influenza viruses, as well as influenza viruses with drug-resistance mutations. The combination of baloxavir with NAIs led to significant synergistic effects; however, the combination of baloxavir with laninamivir failed to result in a synergistic effect on influenza B viruses. Considering the rapid emergence of drug resistance to baloxavir, we believe that these results will be beneficial for combined drug use against influenza.

## 1. Introduction

The influenza virus is a major infectious pathogen, causing acute respiratory disease that leads to extensive medical and economic losses in annual epidemics and periodic pandemics [1,2]. The World Health Organization (WHO) estimates that about 1 billion seasonal flu cases are reported worldwide each year, including 3 million to 5 million severe cases, which cause 290,000–650,000 deaths annually [3,4]. So far, four classes of drugs have been approved for prophylaxis and treatment against influenza virus infections: M2 ion channel blockers, neuraminidase inhibitors, RNA-dependent RNA polymerase (RdRp) inhibitors, and polymerase acidic endonuclease inhibitors [5]. Neuraminidase inhibitors (NAIs) include oseltamivir acid, zanamivir, laninamivir, and peramivir. The ideal bioavailability of oseltamivir makes it highly suitable as an oral formulation, and it is currently the most widely used anti-influenza drug. However, the spread of the drug-resistance mutation to oseltamivir in the influenza A (H1N1) virus has raised great concern among scientists [6]. Zanamivir is a neuraminidase inhibitor effective against influenza A and B viruses that is administered nasally or through oral inhalation, which was approved by the US Food and Drug Administration in 1999 [7]. Laninamivir is a long-acting NAI that is converted to its active form in the inhalation tract after inhalation as a prodrug and is effective against influenza A and B virus strains [8]. Peramivir is a cyclopentane derivative that is structurally different from other approved neuraminidase inhibitors and has preferable in vitro and in vivo efficacies against influenza viruses [9].

Baloxavir marboxil (hereafter referred to as baloxavir) the prodrug of baloxavir acid, is a first-in-class, small-molecule inhibitor of the cap-dependent endonuclease (CEN) reaction that is conducted using the polymerase acidic (PA) protein subunit of the influenza virus polymerase complex [10]. Baloxavir marboxil (baloxavir) was approved as a single-dose treatment for uncomplicated influenza A and B in Japan and the United States in 2018 [11]. It inhibits viral RNA transcription by selectively inhibiting CEN activity in enzyme analysis and inhibits viral replication in infected cells without cytotoxicity in cytopathic analysis. Moreover, baloxavir has antiviral activity against influenza A and B viruses in vitro [12].

Despite the effectiveness of these drugs at reducing influenza-related morbidity and mortality, the emergence of drug-resistant variants poses a critical limitation to their application. Shinya Omoto et al. determined the co-crystal structures of wild-type and I38T endonuclease domains from influenza A and B viruses with bound baloxavir, demonstrating that mutations of I38 in PA, which constitute a major pathway for reduced susceptibility to baloxavir in these variants, only occurred in patients treated with baloxavir [13].

Through the study of Shinya Omoto et al., it was found that PA-I38T substitution is the main way to reduce BXA susceptibility, and the EC_50_ changes of influenza A virus and B virus are 30–50 times and 7 times, respectively [13]. Henju Marjuki and colleagues have shown that oseltamivir treatment was not effective in inhibiting the replication of the R292K mutant strain of the influenza virus in ferrets [14]. Anna Gillman et al. found that the drug sensitivity of OC was reduced by about 13,000 times and that of zanamivir by about 7.8 times against the H6N2 R292K mutant [15]. Dagmara Bialy et al. found that an R292K substitution in H5N6 and H5N2 viruses significantly reduced susceptibility to three licensed NAIs: oseltamivir, zanamivir, and peramivir [16].

Combined drug therapy is a feasible strategy to address the issue of drug resistance. In vitro studies performed by Govorkova et al. demonstrated that the combination of a neuraminidase inhibitor with rimantadine can additively or synergistically inhibit the H1N1 and H3N2 subtypes of influenza A viruses in MDCK cells [17]. Moreover, combination treatment with peramivir and favipiravir had a synergistic effect on DBA/2 mice infected with the oseltamivir-resistant H1N1 virus. This treatment not only significantly reduced the mortality rate among the infected mice but also led to an enhanced antiviral response within the lung tissues, underscoring its potential as an effective strategy against drug-resistant strains [18]. A triple-combination antiviral-drug (TCAD) regimen consisting of amantadine, ribavirin, and oseltamivir is highly synergistic against a wide range of H1N1-resistant and H5N1 virus strains [19]. Peramivir in combination with rimantadine for the treatment of mice infected with the H3N2 virus resulted in synergetic effects on improving mice body weight [20]. Liva Checkmahomed et al. assessed the inhibitory effects of baloxavir acid with three NAIs (oseltamivir, zanamivir, and peramivir) or two other polymerase inhibitors (favipiravir and ribavirin) in cell cultures and in human airway epithelia (HAE) infected with influenza A (H1N1) pdm09 and A (H3N2) viruses. The authors found that the above combinations of baloxavir acid and NAIs or favipiravir had synergistic effects on cell viability against the two influenza A subtypes [21].

However, determining whether combination therapy with baloxavir and NAIs is an effective strategy against influenza drug resistance is a topic in need of further discussion. Here, baloxavir-resistant strains, NAI-resistant strains, and influenza B virus were used for combination studies of baloxavir with four NAIs.

## 2. Materials and Methods

### 2.1. Cells, Virus, and Compounds

Madin–Darby canine kidney (MDCK, ATCC) cells were maintained in Dulbecco’s modified Eagle’s medium (DMEM) (GIBCO BRL, Grand Island, NY, USA) supplemented with 10% fetal bovine serum (FBS) (GIBCO BRL, Grand Island, NY, USA), 100 U/mL of penicillin, and 100 μg/mL of streptomycin at 37 °C and 5% CO_2_. Influenza A virus A/PR/8/34-I38T/R292K mutant strains (PR8-I38T/R292K, H1N1) and A/WSN/33-I38T (WSN-I38T, H1N1) were propagated for 48 h at 37 °C in the chorioallantois cavity of 10-day-old embryonated chicken eggs. Influenza A/Beijing/073103/2009 (H3N2) and influenza B/Lee/40 were propagated in MDCK cells for 3 days at 37 °C in serum-free Dulbecco’s modified Eagle’s medium/Ham’s F-12 medium (DF-12) containing 2 μg/mL tosylamido phenylethyl chloromethyl ketone (TPCK)-treated trypsin (Sigma, Kawasaki, Japan). The virus strains were stored in our lab. The virus yield was determined via plaque assay or an infectivity assay of 50% tissue culture infective dose (TCID_50_). Oseltamivir acid, laninamivir, zanamivir, and baloxavir acid were purchased from MedChemExpress (Shanghai, China) (https://www.medchemexpress.cn), while peramivir trihydrate was purchased from Selleck Chemicals (Shanghai, China) (https://www.selleck.cn).

### 2.2. Combinational Effects of Baloxavir and NAIs

MDCK cells in 96-well plates were infected with influenza virus baloxavir-resistant strains (PR8-I38T, WSN-I38T) at 100 TCID_50_/well, followed by the addition of baloxavir and NAIs in serial dilutions (for baloxavir, 15.63–1000 nM; for oseltamivir acid, 0.63–10 μM; for zanamivir, 0.31–5 μM; for laninamivir and peramivir, 0.06–1 μM). Similarly, a serial dilution of baloxavir and NAIs (for baloxavir, 1.56–100 nM; for NAIs, zanamivir, 0.63–10 μM) was added in a cell assay with influenza virus PR8-R292K mutant strain infected. A serial dilution of baloxavir and NAIs (for baloxavir, 1.56–100 nM; for oseltamivir acid and peramivir, 6.25–100 nM; for zanamivir, 0.63–10 μM; for laninamivir, 0.06–1 μM) was added in a cell assay with influenza A virus H3N2-infected A serial dilution of baloxavir and NAIs (for baloxavir, 1.56–100 nM; for oseltamivir acid, 1.25–20 μM; for zanamivir, 0.63–10 μM; for laninamivir and peramivir, 0.06–1 μM) was added in a cell assay with influenza B virus-infected mice. After incubation at 37 °C for 3 days, the antiviral effect was measured using a CellTiter-Glo cell viability assay (Promega, Madison, WI, USA). Data were analyzed using MacSynergy II software.

### 2.3. Statistical Analysis

The combinational effects were performed using the MacSynergy II program [22]. The three-dimensional peaks above and below the plane were represented as quantitated synergy and antagonism of the interaction of both indicated drugs, respectively. The plane showed the theoretical additive effect of both drugs, which was calculated via dose response curves for individual drugs and was defined as either having no synergy or antagonism. The values of synergy or antagonism volumes under 25 μM^2^% were regarded as insignificant, while those between 25 μM^2^% and 50 μM^2^% were regarded as minor, between 50 μM^2^% and 100 μM^2^% as moderate, and over 100 μM^2^% as strong.

## 3. Results

### 3.1. Baloxavir and NAIs Have Synergistic Antiviral Effects against Baloxavir-Resistant Influenza Strains In Vitro

In the previous study, the investigated influenza drug-resistant strains were constructed, and the antiviral activities of baloxavir and NAIs against these resistant strains were listed in Appendix A. Firstly, we evaluated the synergetic effects of baloxavir and NAIs using an 8 × 6 combinatorial design by monitoring CPE in MDCK cells. Baloxavir was combined with NAIs (oseltamivir acid, zanamivir, laninamivir, and peramivir) to treat MDCK cells infected with the H1N1-PR8-I38T mutant, which was resistant to baloxavir. The data were plotted and analyzed using the MacSynergy II software. The results showed that baloxavir and NAIs significantly prevented PR8-I38T infection-induced CPE in a dose-dependent manner (Figure 1). For the combination of baloxavir with NAIs, the volume of synergy was 288.54 µM^2^%, 224.19 µM^2^%, 294.33 µM^2^%, and 180.47 µM^2^%, respectively, at 95% confidence levels, which is interpreted as representing a strong synergy effect against PR8-I38T in vitro (Table 1).

Then, we measured the interactions between baloxavir in combination with NAIs to treat MDCK cells infected by H1N1-WSN-I38T. The data were plotted and analyzed using the MacSynergy II software. The results showed that baloxavir combined with oseltamivir or zanamivir significantly prevented WSN-I38T infection-induced CPE in a dose-dependent manner (Figure 2A,B). Baloxavir combined with laninamivir or peramivir slightly prevented H1N1-WSN-I38T-induced CPE (Figure 2C,D). For the combination of baloxavir with oseltamivir acid or zanamivir, the volume of synergy was 311.56 µM^2^% or 394.35 µM^2^%, respectively, at 95% confidence levels, which suggested strong synergy effects against H1N1-WSN-I38T in vitro. For the combination of baloxavir with laninamivir or peramivir, the volume of synergy was 52.02 µM^2^% or 88.70 µM^2^%, respectively, at 95% confidence levels, which would be defined as a moderate synergy effect against H1N1-WSN-I38T in vitro (Table 2). Taken together, the co-administration of baloxavir and NAIs exhibited synergistic effects in suppressing baloxavir-resistant strains.

### 3.2. Baloxavir and NAIs Have Synergistic Antiviral Effects against NAI-Resistant Strains In Vitro

Oseltamivir acid, zanamivir, laninamivir, and peramivir do not inhibit H1N1-PR8-R292K infection-induced CPE in MDCK cells. Therefore, we evaluated the synergetic effects of baloxavir and NAIs using an 8 × 6 combinatorial design by monitoring CPE in MDCK cells. The data were plotted and analyzed using the MacSynergy II software. The results showed that baloxavir combined with oseltamivir acid or peramivir significantly prevented H1N1-PR8-R292K infection-induced CPE in a dose-dependent manner (Figure 3A,D). Baloxavir combined with laninamivir or zanamivir slightly prevented H1N1-PR8-R292K infection-induced CPE (Figure 3B,C). For the combination of baloxavir with oseltamivir acid or peramivir, the volume of synergy was 287.87 µM^2^% or 138.87 µM^2^%, respectively, at 95% confidence levels, which suggested strong synergy effects against H1N1-PR8-R292K in vitro. For the combination of baloxavir with laninamivir or zanamivir, the volume of synergy was 72.3 µM^2^% or 93.35 µM^2^%, respectively, at 95% confidence levels, which is defined as a moderate synergy effect against H1N1-PR8-R292K in vitro (Table 3). In summary, the co-administration of baloxavir and NAIs exhibited synergistic effects in suppressing NAI-resistant strains.

### 3.3. Baloxavir and NAIs Have Synergistically Antiviral Effects on Influenza B and Influenza A H3N2 In Vitro

Finally, we examined the inhibitory effects of the combination of baloxavir and NAIs on influenza A and B viruses in vitro using an 8 × 6 combinatorial design by monitoring CPE in MDCK cells. The data were plotted and analyzed using the MacSynergy II software. The results indicate that baloxavir combined with zanamivir significantly prevented influenza B virus infection-induced CPE in a dose-dependent manner (Figure 4C). While baloxavir combined with oseltamivir acid or peramivir slightly prevented influenza B virus-induced CPE (Figure 4A,D), baloxavir combined with laninamivir did not prevent influenza B virus-induced CPE (Figure 4B). For the combination of baloxavir with zanamivir, the volume of synergy was 137.14 µM^2^%, respectively, at 95% confidence levels, which suggested strong synergy effects against influenza B virus in vitro. On the other hand, the combination of baloxavir with oseltamivir acid or peramivir showed a synergy volume of 76.70 µM^2^% or 82.03 µM^2^%, respectively, at 95% confidence levels, which is defined as a moderate synergy effect against influenza B virus in vitro (Table 4). Moreover, we found that baloxavir combined with oseltamivir acid, laninamivir, and zanamivir significantly prevented H3N2 infection-induced CPE in a dose-dependent manner (Figure 5A–C), whereas baloxavir combined with peramivir slightly prevented H3N2-induced CPE (Figure 5D). For the combination of baloxavir with oseltamivir acid, laninamivir, and zanamivir, the volume of synergy was 191.02 µM^2^%, 268.57 µM^2^%, or 155.61 µM^2^%, respectively, at 95% confidence levels, which suggested strong synergy effects against H3N2 in vitro. For the combination of baloxavir with peramivir, the volume of synergy was 97.97 µM^2^%, respectively, at 95% confidence levels, which would be defined as a moderate synergy effect against H3N2 in vitro (Table 5). These results indicate that the co-administration of baloxavir and NAIs exhibited synergistic effects in suppressing influenza B and A viruses.

## 4. Discussion

Baloxavir and NAIs are FDA-approved drugs for the treatment of influenza virus infection. However, drug-resistant mutants can arise either naturally or due to drug stress caused by the clinical use of baloxavir and NAIs. Therefore, it is important to identify other treatment options in the event of drug-resistant virus variants. Combination therapy with two or more approved antiviral drugs is considered a potential option for influenza treatment. However, there is limited data regarding combination therapy with two or more antivirals to combat drug-resistant strains. In this study, we selected major first-line anti-influenza drugs, baloxavir and NAIs, and systematically explored the preferred drug combinations against different resistant strains.

Influenza A viruses have been shown to develop resistance to baloxavir by acquiring substitutions at one of the highly conserved residues in the PA catalytic site, isoleucine 38 (I38X) [23]. Chesnokov et al. found that viruses bearing I38T and I38S mutations were less susceptible to baloxavir [24]. Previous studies have shown that viruses with the NA-R152K and PA-I38T double mutations have reduced susceptibility to NA inhibitors and baloxavir in vivo and in vitro, suggesting that this antiviral may be ineffective in people infected with this double-mutant virus [25]. R292 is one of three key conserved arginine residues in the active site that surrounds the carboxylate group of sialic acid. Mtambo et al. found that the R292K mutation in H7N9 neuraminidase decreased the binding with peramivir [26]. In addition, some investigators have found that the R292K mutation highly reduces susceptibility to oseltamivir as well as to zanamivir and peramivir, but mostly retains susceptibility to non-NAIs (favipiravir, nitazoxanide, and ribavirin) [27]. Due to the emergence of resistance mutations in influenza viruses, the effectiveness of existing drugs has diminished, and, therefore, there is an urgent need to develop drug combination strategies against drug-resistant strains before a novel anti-influenza drug is approved.

The researchers found that the combination of oseltamivir and ribavirin was more effective than the use of either one alone in terms of antiviral activity against H5N1, both in vitro and in vivo [28]. Hoopes et al. found a sustained suppression of drug-resistant influenza viruses using a TCAD regimen, which was superior to dual combinations and single drugs in suppressing resistance, and that at least three drugs were needed to block the selection of drug-resistant variants of the influenza A virus [29]. From these findings, we conclude that the use of combination regimens is more effective in countering the epidemic of influenza virus resistance. In this study, we demonstrated the antiviral effect of the combination of baloxavir and NAIs against drug-resistant strains of influenza viruses, including influenza A and B viruses, in vitro, and outlined the scheme of anti-influenza drug combinations against different resistant strains.

## 5. Conclusions

Liva Checkmahomed et al. demonstrated that the combination of BXA and NAIs or favipiravir had a synergistic effect on the cell viability of two influenza A virus subtypes. On the other hand, the combined application of BXA and ribavirin showed different results [21]. Therefore, in this study, we performed additional in vitro studies using resistant strains (I38T or R292K variants) to evaluate the effect of combination therapy. Our study demonstrated that the combination of baloxavir and NAIs has a strong synergistic effect on PR8-I38T in vitro. The combination of baloxavir with oseltamivir or zanamivir had a strong synergistic effect on WSN-I38T in vitro, while its combination with laninamivir or peramivir had a relatively weak synergistic effect. The combination of baloxavir with oseltamivir or peramivir had a strong synergistic effect on PR8-R292K in vitro, while its combination with laninamivir or zanamivir had a weak synergistic effect. The combination of baloxavir and zanamivir has a strong synergistic effect on influenza B virus in vitro, while its combination with oseltamivir or peramivir had a weak synergistic effect. However, the combination of baloxavir and laninamivir was not effective for influenza B. The combination of baloxavir with NAIs had a strong synergistic effect on H3N2 in vitro, and its combination with peramivir had a weak synergistic effect. Overall, the combinations of baloxavir and most NAIs presented synergistic effects against both the resistant strains and susceptible strains of influenza viruses in this study, and the synergistic use had a stronger effect on drug-resistant strains, especially on the baloxavir (I38T)-resistant strain. In particular, the drug combinations of oseltamivir and baloxavir showed high or moderate synergistic effects on multiple influenza virus strains investigated. We believe that these results will be helpful to future combat against influenza drug resistance and guide future clinical medication.

## Figures and Tables

**Figure 1 viruses-16-01467-f001:**
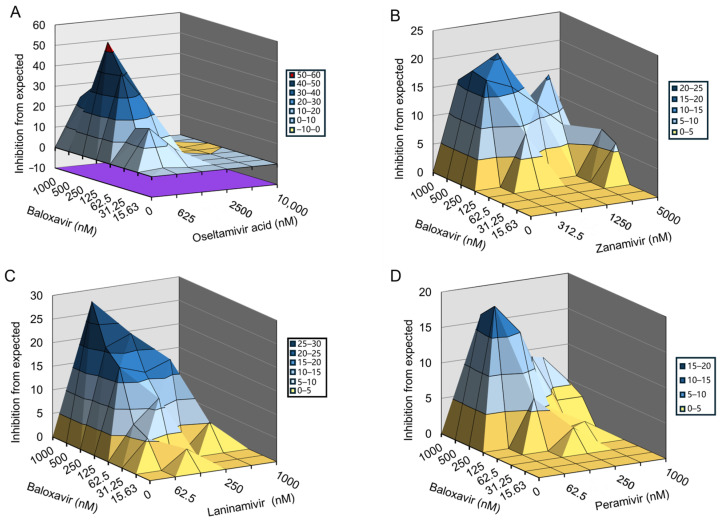
Inhibitory effect of baloxavir in combination with NAIs in cells infected with H1N1-PR8-I38T mutant strain. The antiviral activity of baloxavir + NAIs was evaluated against H1N1-PR8-I38T in MDCK cells. Graphs showing the antiviral activity of baloxavir in combination with (**A**) oseltamivir acid; (**B**) zanamivir; (**C**) laninamivir; and (**D**) peramivir.

**Figure 2 viruses-16-01467-f002:**
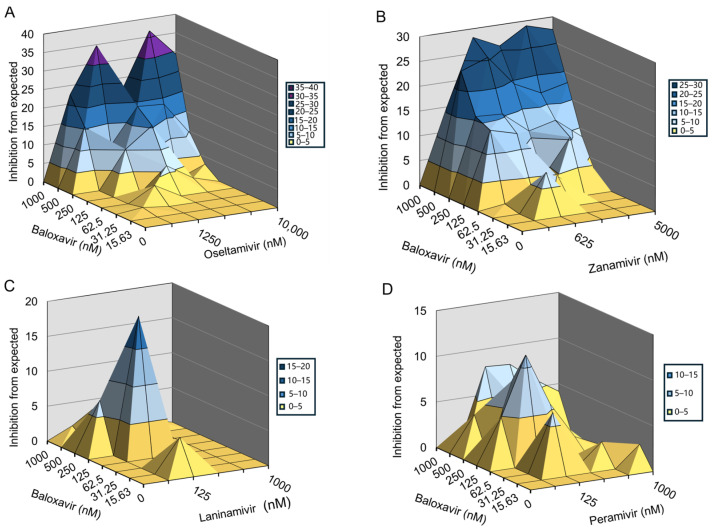
Inhibitory effect of baloxavir in combination with NAIs in cells infected with H1N1-WSN-I38T mutant strain. The antiviral activity of baloxavir + NAIs was evaluated against H1N1-WSN-I38T in MDCK cells. Graphs showing the antiviral activity of baloxavir in combination with (**A**) oseltamivir acid; (**B**) zanamivir; (**C**) laninamivir; and (**D**) peramivir.

**Figure 3 viruses-16-01467-f003:**
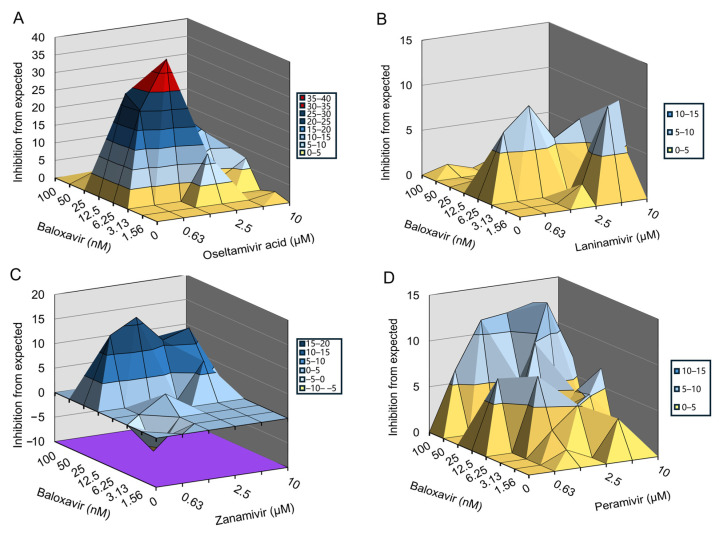
Inhibitory effect of baloxavir in combination with NAIs in cells infected with H1N1-PR8-R292K mutant strain. The antiviral activity of baloxavir + NAIs was evaluated against H1N1-PR8-R292K in MDCK cells. Graphs showing the antiviral activity of baloxavir in combination with (**A**) oseltamivir acid; (**B**) laninamivir; (**C**) zanamivir; and (**D**) peramivir.

**Figure 4 viruses-16-01467-f004:**
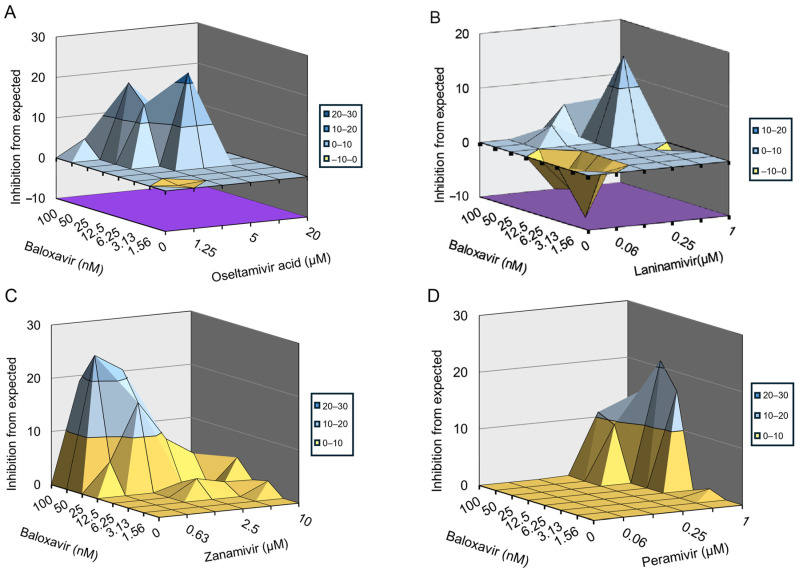
Inhibitory effect of baloxavir in combination with NAIs in cells infected with influenza B strain. The antiviral activity of baloxavir + NAIs was evaluated against the influenza B strain in MDCK cells. Graphs showing the antiviral activity of baloxavir in combination with (**A**) oseltamivir acid; (**B**) laninamivir; (**C**) zanamivir; and (**D**) peramivir.

**Figure 5 viruses-16-01467-f005:**
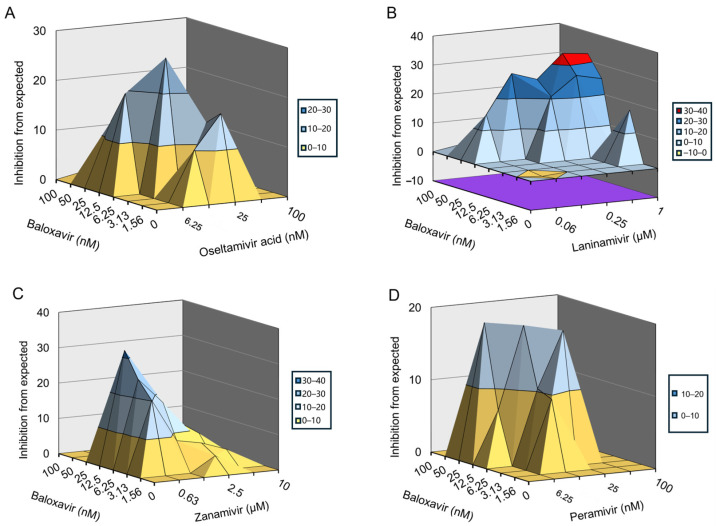
Inhibitory effect of baloxavir in combination with NAIs in cells infected with influenza H3N2. The antiviral activity of baloxavir + NAIs was evaluated against influenza H3N2 in MDCK cells. Graphs showing the antiviral activity of baloxavir in combination with (**A**) oseltamivir acid; (**B**) laninamivir; (**C**) zanamivir; and (**D**) peramivir.

**Table 1 viruses-16-01467-t001:** Interactions of drug–drug combinations against influenza H1N1-PR8-I38T mutant strain.

Drug Combination	Synergy/Antagonism (μM^2^%) *	Predicted Interaction
Baloxavir + Oseltamivir acid	288.54/−1.03	Strong synergy
Baloxavir + Zanamivir	224.19/0	Strong synergy
Baloxavir + Laninamivir	294.33/0	Strong synergy
Baloxavir + Peramivir	180.47/0	Strong synergy

* Mean volumes of synergy or antagonism are presented based on 95% confidence levels.

**Table 2 viruses-16-01467-t002:** Interactions of drug–drug combinations against influenza H1N1-WSN-I38T mutant strain.

Drug Combination	Synergy/Antagonism (μM^2^%) *	Predicted Interaction
Baloxavir + Oseltamivir acid	311.56/0	Strong synergy
Baloxavir + Zanamivir	394.35/0	Strong synergy
Baloxavir + Laninamivir	52.02/0	Moderate synergy
Baloxavir + Peramivir	88.70/0	Moderate synergy

* Mean volumes of synergy or antagonism are presented based on 95% confidence levels.

**Table 3 viruses-16-01467-t003:** Interactions of drug–drug combinations against influenza H1N1-PR8-R292K mutant strain.

Drug Combination	Synergy/Antagonism (μM^2^%) *	Predicted Interaction
Baloxavir + Oseltamivir acid	287.87/0	Strong synergy
Baloxavir + Laninamivir	72.3/0	Moderate synergy
Baloxavir + Zanamivir	93.35/−7.8	Moderate synergy
Baloxavir + Peramivir	138.87/0	Strong synergy

* Mean volumes of synergy or antagonism are presented based on 95% confidence levels.

**Table 4 viruses-16-01467-t004:** Interactions of drug–drug combinations against influenza B virus.

Drug Combination	Synergy/Antagonism (μM^2^%) *	Predicted Interaction
Baloxavir + Oseltamivir acid	76.70/−0.52	Moderate synergy
Baloxavir + Laninamivir	33.21/−27.12	
Baloxavir + Zanamivir	137.14/0	Strong synergy
Baloxavir + Peramivir	82.03/0	Moderate synergy

* Mean volumes of synergy or antagonism are presented based on 95% confidence levels.

**Table 5 viruses-16-01467-t005:** Interactions of drug–drug combinations against influenza H3N2.

Drug Combination	Synergy/Antagonism (μM^2^%) *	Predicted Interaction
Baloxavir + Oseltamivir acid	191.02/0	Strong synergy
Baloxavir + Laninamivir	268.57/−0.66	Strong synergy
Baloxavir + Zanamivir	155.61/0	Strong synergy
Baloxavir + Peramivir	97.97/0	Moderate synergy

* Mean volumes of synergy or antagonism are presented based on 95% confidence levels.

## Data Availability

Data are contained within the article.

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
