# Peer review of "The Synergistic Effect of Baloxavir and Neuraminidase Inhibitors against Influenza Viruses In Vitro"

_viruses, 2024, doi:10.3390/v16091467_

Round 1

Reviewer 1 Report

Comments and Suggestions for Authors

Guo and colleagues prepared a manuscript on the synergistic effect of baloxavir and neuraminidase inhibitors against influenza viruses in vitro. As follows from the abstract, the combination of baloxavir with NAIs led to significant synergistic effects; while, the combination of baloxavir with laninamivir failed to result in a synergistic effect on influenza B viruses. The manuscript contains enough information indicating the importance of the study, however, I have a number of comments, after which the article can be published.

The title of the manuscript should not contain abbreviations and acronyms, so the full name neuraminidase inhibitors should be given

In the abstract, the decoding of neuraminidase inhibitors should be given at the first mention

The introduction is written quite clearly, but some sentences are not clear from the point of view of the English language. For example, In 2018, baloxavir was approved for influenza A and B [10].

The manuscript content needs to be checked by a native English speaker to improve the quality of the article

In addition, the manuscript, in particular the discussion of the results, contains some grammatical and technical errors, such as incorrect sentence hyphenation, dashes, which should be eliminated

The discussion and the main conclusion need to be improved by adding the authors' assumptions about why such results were obtained, supporting their conclusions with literary references if possible. In addition, the conclusion should contain some generalization of the results, as well as prospects for the results obtained for further application.

Reviewer 2 Report

Comments and Suggestions for Authors

The manuscript “The synergistic effect of baloxavir and NAIs against influenza viruses in vitro” is devoted to the assessment of effect of combined use of anti-influenza drugs of different mechanisms of activity. Main characteristics of synergy regarding baloxavir- and NAI-resistant strains have been determined in the course of the study.

The manuscript should be seriously improved prior to publication. The main drawback is the lack of principal characteristics of viruses used. The only results of the study are values of synergistic effect for various combinations of antivirals. Authors postulate that they used baloxavir- and NAI-resistant viruses. However, no indications are presented to resistance or susceptibility of viruses to specific drug, viruses are marked by amino acid substitutions only. Is PR8-R292K virus resistant to baloxavir? Resistant to NAI? Susceptible to both? If resistant to NAI, what specific drug(s) of NAI class it is resistant to? The same questions should be answered about all other viruses used in the study.

Why authors selected influenza B virus for the study? What are its main differences from other viruses? Is it susceptible or resistant to the drugs under investigation?

How was the baloxavir- and NAI-resistance confirmed phenotypically? What is the value of IC50’s for sensitive viruses and mutants used in the study against the drugs used?

The extent of synergy regarding combination of baloxavir with different NAI’s appeared different. How authors explain this?

Stronger conclusion for practical use of antivirals should be done. For instance, what is optimal combination of drugs?

Minor changes.

p.1. Change to read “four classes of drugs

p.2. “…protected them from severe pathogenesis”. Please re-phrase to avoid protection from pathogenesis, not disease.

Page 3, first line. Change to read “…with influenza virus baloxavir-resistant…”

Page 8. Change to read “…found that viruses bearing I38T and I38S mutations were less sensitive…”

Comments on the Quality of English Language

p.1. Change to read “four classes of drugs

p.2. “…protected them from severe pathogenesis”. Please re-phrase to avoid protection from pathogenesis, not disease.

Page 3, first line. Change to read “…with influenza virus baloxavir-resistant…”

Page 8. Change to read “…found that viruses bearing I38T and I38S mutations were less sensitive…”
